# Translational Strategies to Target Metastatic Bone Disease

**DOI:** 10.3390/cells11081309

**Published:** 2022-04-12

**Authors:** Gabriel M. Pagnotti, Trupti Trivedi, Khalid S. Mohammad

**Affiliations:** 1Department of Endocrine, Neoplasia and Hormonal Disorders, MD Anderson Cancer Center, University of Texas, Houston, TX 77030, USA; gmpagnotti@mdanderson.org (G.M.P.); ttrivedi@mdanderson.org (T.T.); 2Department of Anatomy and Genetics, Alfaisal University, Riyadh 11533, Saudi Arabia

**Keywords:** metastatic bone disease, osteolytic lesions, osteoblastic lesions, bone marrow microenvironment, radiotherapy, palliative therapy, immunotherapy

## Abstract

Metastatic bone disease is a common and devastating complication to cancer, confounding treatments and recovery efforts and presenting a significant barrier to de-escalating the adverse outcomes associated with disease progression. Despite significant advances in the field, bone metastases remain presently incurable and contribute heavily to cancer-associated morbidity and mortality. Mechanisms associated with metastatic bone disease perpetuation and paralleled disruption of bone remodeling are highlighted to convey how they provide the foundation for therapeutic targets to stem disease escalation. The focus of this review aims to describe the preclinical modeling and diagnostic evaluation of metastatic bone disease as well as discuss the range of therapeutic modalities used clinically and how they may impact skeletal tissue.

## 1. Clinical Presentation of Metastatic Bone Disease

Bone is the third most common site of metastasis, the progression of which contributes significantly to mortality [1]. Management of primary solid tumors can be complicated when coupled with tumors established in the bone marrow microenvironment, creating unique clinical challenges. Unfortunately, evidence of the primary disease usually becomes evident only once bone metastases are discovered, typically triggered following complaints of bone pain [2] or pathological fracture [3] (Figure 1). Approximately 70% of cancers that progress into metastatic bone tumors are derived from breast, prostate, and lung cancers, with other cancers (e.g., medullary and anaplastic thyroid, renal, gynecologic, melanoma, and gastric carcinomas) contributing to a range of varying metastatic frequency [4,5]. Efforts to reach, contain and treat the disease become progressively difficult, signifying tipping points for poor prognoses [6]. As skeletal metastases are largely incurable, the severity of the altered bone phenotype to favor tumor cell colonization leads to increased patient mortality, which is derived from an increase in skeletal-related events (SRE) [7,8], resistance to treatments and related morbidities. Even though conventional treatment approaches have been standardized and complimented by bone-targeted therapies for increased precision, five-year survival rates for bone metastases do not currently exceed 20%. 

The metastatic event is a product of primary tumor cell dissemination and communication with a distant tissue harboring favorable growth dynamics [9]. High vascularity to bone mediates access to niches bearing trophic and growth factors suspended within the marrow and inflammatory cytokines [10] that accelerate tumor colonization embedded within the bone matrix, a postulate proposed and widely regarded as the “seed-soil” hypothesis by Paget [11]. Circulating tumor cells constitute the “seed”, while the bone and marrow provide the “congenial soil” rife with inflammatory mediators, hormones, and bone-derived factors, including transforming growth factor-β (TGF-β), for growth and proliferative cues [12]. Uncoupling and rewiring the established, tightly-regulated marrow microenvironment into a pre-metastatic niche accommodates the colonizing cells [13]. Interceding tumor cells disrupt bone homeostasis orchestrated by resident bone remodeling cells to advantageously favor either or both resorption and formation processes. Established “vicious cycles” of osteolytic, osteoblastic, or mixed lesions ultimately undermine the structural and mechanical integrity of bone and degrade marrow health [14,15] (Figure 1).

Given the range and frequency of primary solid tumors that progress into bone metastases, entire fields are dedicated to elucidating the nuances associated with their mechanisms of action, detection, and individualized treatment sensitivities. For instance, breast cancer metastases detected by conventional radiography, unless abnormal radionuclide uptake necessitates PET/CT imaging [16], are predominantly osteolytic lesions driven by heightened osteoclast resorption that outpaces bone formation. Similarly, renal cell carcinomas that metastasize to the bone present as osteolytic lesions in the pelvis, ribs, and spine [17], Conversely, prostate cancer bone metastases exhibit sclerotic bone upon radiographic analysis, suggesting an overtly osteoblastic process (though resorption plays a role as well) [18] fueled by bone morphogenic protein (BMP), epidermal growth factor (EGF), and platelet-derived growth factor (PDGF) [19,20]. Stimulated tumor cells produce inflammatory cytokines that act in autocrine fashion to increase tumor growth and in induce bone lesions. Primary lung cancers are subject to screening and staging regarding preoperative imaging of bone [21]. Stage III and IV patients harboring bone metastases with tumors of differing phenotypes, exhibit distinct radiologic appearance upon extra-thoracic metastasis to bone; more significant cases show osteolytic lesions (i.e., pulmonary adenocarcinomas, non-small cell lung cancer) [22], whereas other subtypes present osteoblastic lesions (i.e., small cell) often found in appendicular bone (scaphoid and phalanges) [23]. Neuroendocrine thyroid cancers present equally variable degrees of bone metastases. For instance, slow-growing differentiated Hurthle cell, papillary, and follicular thyroid cancers account for low rates of bone metastasis, in contrast to medullary and anaplastic thyroid cancers, which have higher rates of metastasis and equally poorer prognoses [24]. 

The ensuing treatment strategies are dictated by setting and disease profile, responsivity to other treatments, and are further compounded by patient age and health [25]. Furthermore, the potential adverse effects on musculoskeletal and systemic endpoints should always be taken into consideration. Yet, despite advances in drug discovery and cancer therapeutics and our evolving understanding of disease mechanisms, bone metastases remain largely incurable. Efficacious cancer treatments notwithstanding, cancers that have metastasized to the bone remain challenging in their ability to reemerge after near eradication of the disease in addition to the secondary side effects, particularly on bone and muscle. These effects may accelerate pre-menopausal osteoporosis and lead to cancer treatment-associated bone loss and cachexia through distinct pathways resulting from the disease or treatment. For example, ovarian tissue in women is the primary source of estrogen, a dominant regulator of bone [26]. Since the ovaries are sensitive to chemotherapy, repeated exposure can profoundly suppress estrogen synthesis [27] and impair other ovarian functions, such as reproduction [28]. Changes in estrogen levels or its substrates can disrupt and undermine the downstream pathways dependent on estrogen, especially those regulating bone health [29]. Compounding these adverse effects with age-related osteoporosis can accelerate bone loss [30] and susceptibility to SRE. Muscle also responds poorly to cancer therapies, inducing a cachexic state that leads to muscle wasting and weakness [31].

Bone marrow is a metabolically active tissue housing stem cells of mesenchymal and hematopoietic origin while also functioning to coordinate the bone modeling and remodeling activities between a multitude of bone and immune cells, all of which maintain the integrity of bone health and the immune system. Thus, the bone and the accompanying marrow, are highly responsive to changes in metabolism. Precise orchestration of the remodeling process within the bone microenvironment is physically disrupted by the imposition of bone metastases, outcompeting healthy resident cells for space. Production of this increasingly tumor-supportive, pro-inflammatory niche correlates to drastic declines in bone health. Therefore, the omission of key cells and molecules involved in bone remodeling and hematopoiesis imparts adverse downstream consequences on patient health. Elucidating these pathways has facilitated the development of agents that target elements of the bone remodeling pathway as well as others which exclusively target the tumor. 

## 2. Cellular Mechanisms Guiding Bone Metastases

Cancer cells migrate with high affinity to marrow spaces and develop into skeletal metastases. Comprehensive reasoning behind how and why this is achieved is still under investigation, although much research has been accomplished in the field in recent decades. Increasing evidence has elucidated the bone tropism of cancer cells. Adopting the notion of a pre-metastatic niche has been described as how primary tumor cells conspire with autocrine-secreted factors and distant stromal cells to reprogram the bone microenvironment for colonization [32,33]. Key features driving this event include heightened inflammation, low-oxygen tension (hypoxia), access to growth factors, and a durable vascular network [34]. Priming the pre-metastatic niche via exosomes has been described as a precursor event to tumor homing and colonization [35,36] as well. Fibroblasts and osteoblast-lineage cells, already residing within the marrow microenvironment [37], also contribute to homing of the cancer cells to bone [38]. Connexin-43 gap junctions have been specifically implicated in breast cancer bone metastases and are inversely correlated to patient survival [39]. 

Once tumor cells invade the marrow, inflammatory cytokines combined with the low-oxygen tension environment are highly favorable and drive tumor cell integration and growth. Bone metastases are diagnosed radiographically as either osteolytic or osteoblastic. The mechanisms driving the activity of bone resorption and/or formation determine the appearance of the bone lesions on X-ray. Considering the dramatic imbalances in the bone due to bone metastases, bone remodeling cells have been the target of many therapeutic interventions. However, tumor-secreted factors and local pro-inflammatory cytokines play a significant role in this orchestrated transformation. Evidence has shown that tumors thriving in a low oxygen environment [40] are more resistant to chemotherapies and radiotherapies [41]. 

Maturation and the resorptive activity of osteoclasts are dependent on and are highly upregulated when bound to rate-limiting cytokines macrophage colony-stimulating factor (MCSF) and receptor activator of nuclear factor Kappa-Beta ligand (RANKL). Inflammation is favorable to metastatic tumors, facilitating osteoclast contributions to tumor progression [15]. For instance, parathyroid hormone-related peptide (PTHrP), a factor secreted by tumor cells, plays a significant role in the conversion of the bone marrow into that of a hypercalcemic [42] tumor microenvironment [43,44]. The TGF-β superfamily contains key, yet complex mediators of many processes, typically those that function in conjunction with hypoxia to upregulate and sustain tumor growth. Due to the rich concentrations of TGF-β embedded throughout the bone matrix, heightened resorption releases these and other factors, which enhance tumor growth, only perpetuating further osteolytic destruction [45]. Interleukins (i.e., IL-6, IL-11) also play an essential role in mediating osteoclast activity and, together with bone-derived TGF-β, drive further tumor growth and accelerate bone resorption. Recently, more attention has been given to the molecular crosstalk between bone and muscle. While the bulk of osteolytic bone destruction results in increased skeletal fragility, the secondary release of sequestered bone-derived TGF-β has been shown to compromise Ca^2+^ signaling channels in muscle fibers. Degraded voltage potentials result in decreased muscle contractility and reduced muscle strength [46]. Together with poor bone composition, the physiologic changes to muscle reduce quality-of-life. The mechanisms guiding cancer cell affinity to bone [47] may offer the means to deter the onset of progress of established metastatic disease.

## 3. Cancer-Treatment Induced Bone Loss

For hormone-receptor-positive tumors of the breast and prostate, existing treatment strategies target the blockade or hindered synthesis of sex steroids androgen and estrogen for prostate and breast cancers, respectively. Despite the positive effects on reducing tumor burden, these approaches exhibit adverse effects on the musculoskeletal system that can further degrade quality-of-life (Figure 1). Therapeutic modalities and their effect(s) on bone and off-target tissues and organ systems are summarized in Appendix A.

### 3.1. Androgen Deprivation Therapy

Prostate cancer is the leading non-skin cancer in men, with advanced-stage disease leading to exceedingly poor prognoses. Prostate cancer progression depends on the binding between the androgen receptor and testosterone (dihydrotestosterone), which suggests a therapeutic target. Androgen deprivation therapy is highly effective in early-stage prostate cancer and is efficacious to a lesser extent in advanced-stage disease [48]. This can be achieved by surgically- or medically-induced castration, providing near elimination of testosterone synthesis to blunt the advancement of prostate cancer or prostate cancer bone metastases [49]. Surgical castration of the testes and epididymis is achieved via bilateral orchiectomy or clamp ablation [50], with the former exhibiting psychological drawbacks and effective in reducing serum-testosterone and suppressing disease [51]. Intermittent androgen deprivation therapy has shown to be more effective when combined with improved diet, exercise, and vigilance. Alternatively, medical castration is a pharmacological means to lower testosterone without surgical intervention, whether through luteinizing hormone receptor hormone (LHRH) agonists, estrogen, or progesterone. LHRH antagonists bind to their complementary LHRH receptors on pituitary gonadotropin-producing cells. Abarelix was the first US Food and Drug Administration (FDA)-approved LHRH antagonist for advanced prostate cancer. Chemical castration is performed using the drug medroxyprogesterone acetate, a synthetic analog to female-derived progesterone, while also used as a female birth control to inhibit ovulation, is as applicable in suppressing androgen production and, thus, testosterone. Androgen inhibitors, drugs that block enzymes that synthesize testosterone, such as enzalutamide [52], ketoconazole, and abiraterone, have been used as single agents or in combination with steroids in the treatment of metastatic castrate-resistant prostate cancers [53].

Complications arise with these approaches, derived primarily from reduced hormone bioavailability. Contrary to LHRH agonists, testosterone flare is not observed with LHRH antagonists as they do not cause initial release of LH or Follicle-stimulating hormone (FSH). Other physiological considerations are noted as secondary to treatment, such as accelerated osteopenia, altered metabolism, and cognitive decline. Perhaps the most debilitating side effect, especially in the context of this review, is the reduction in testosterone, the molecular substrate for estrogen synthesis. Orchiectomy results in loss of testosterone, yet, castration-resistance is often observed in advanced disease, a mechanism suggested to derive from overexpression of transcriptional intermediary factor 2 (TIF2) and steroid-receptor coactivator 1 (SRC1) [54]. Thus, dramatic decreases in bone mineral density are typically observed following androgen deprivation therapy. Surgical and psychological costs and continuance of medical castration drugs present additional limitations.

### 3.2. Estrogen Deprivation Therapy

Modulating the receptor-binding characteristics involved in the tumor proliferation that responds to estrogen activity or the bioavailability of estrogen itself are strategies used to inhibit breast cancer and its metastases. Managing the care of patients treated with breast cancer begins with considering the hormone status. Approximately 70% of all breast cancers express estrogen-receptors rendering their progression susceptible to downregulated levels of circulating estrogen levels or their substrates. Selective estrogen receptor modulators (SERMs) (i.e., raloxifene, toremifene, and tamoxifen) are unique molecules that bind estrogen receptors (ER) although lack the steroidal component found in estrogens. Their binding characteristics allow selective function as estrogen receptor agonist or antagonist [55]. SERMs are commonly prescribed in younger postmenopausal women to treat osteoporosis and associated fractures and have been efficacious in blunting the progression of estrogen-receptor-positive (ER^+^) breast cancer bone metastases. In recent years, 3rd generation SERM bazedoxifene in combination with palbociclib, a selective cyclin-dependent kinase 4/6 (CDK4/6 inhibitor), has shown first evidence of a CDK inhibitor treating HR^+^ advanced breast cancer [56,57]. As an alternative to SERMs, aromatase inhibitors (AI) (i.e., letrozole) are used predominantly in post-menopausal women with ER^+^ breast cancer. AIs are a class of drugs designed to interrupt estrogen synthesis, which elicits a profound effect on osteoclast regulation. The mechanism of action is rooted in the ability of AI to block the synthesis of the enzyme aromatase, which converts peripheral androgens into estrogens. Complete estrogen deprivation is a clinical standard for ER^+^ breast cancer, significantly reducing mortality rates. In recent years, the use of CDK4/6 inhibitors palbociclib or ribociclib in combination with letrozole have gained FDA approval to treat HR^+^ breast cancer patients [58]. However, considering post-menopausal status parallels an increased likelihood of osteoporosis, depleting the body of critical regulators of osteoclast-mediated resorption can escalate skeletal fragility. In estrogen’s absence, a severe increase in osteoclast activity is likely to be observed, only further perpetuating a patient’s risk for fracture. Additionally, heightened release of TGF-β causes maladaptive modifications to the calcium (Ca^2+^)-channel regulators, the ryanodine receptors. Ultimately, Ca^2+^ leakage across these channels reduces membrane voltage potential and decreases muscle strength [59]. Patients then find themselves at risk for fracture development as well as systemically weak. Myopathies are emerging beyond skeletal muscle weakness, as cardiotoxicity is often reported.

## 4. Modeling, Imaging, and Detection of Skeletal Metastases

### 4.1. Preclinical Modeling

Preclinical research data has provided extensive mechanistic and target-based information in bone metastasis and subsequent treatments. Modeling bone metastases in small animals can be performed using diverse mouse strains and employing various tumor inoculation techniques [60]. Inoculation of MDA-MB-231 or MCF-7 human breast cancer [61] or PC3 and LNCaP human prostate cancer cell lines [62] locally to bone (intra-tibial) or systemically to the left cardiac ventricle (intra-cardiac) are powerful tools to study these diseases in the bone. Intra-tibial inoculation (5 × 10^3^ cells) delivers a precise tumor-cell bolus into the marrow of pre-selected limb bones. The tumor burden’s is thereby confined to a single limb and/or one tissue, providing the researcher with the ability to study isolated effects without systemic disease. Alternatively, the systemic disease can be modeled by intravenous inoculation of tumor cells (1 × 10^5^–2 × 10^6^ cells) into the circulation via the tail vein or through intracardiac inoculation, both of which effectively engender the bones with aggressive metastases. Estrogen deprivation therapy can be modeled in mice by performing ovariectomy in combination with an AI, leading to extensive bone loss [59] and muscle weakness, reflective of AI-treated breast cancer patients [63]. Similarly, modeling androgen deprivation in mice can be achieved through surgical (orchiectomy) and pharmaceutical castration of mice, leading to heightened bone resorption and adverse effects on muscle [64]. Orthotopic tumor inoculation using aggressive cancer cells can be alternatively utilized to develop a systemic model of bone metastasis. For example, breast cancer cell inoculation in the fourth mammary fat pad can develop into aggressive bone metastasis, similar to the natural course of breast cancer metastasis [61]. Spontaneous metastatic tumor models are not common in rodents, although they can occur in canine models bearing prostate cancer and others [65] (Figure 1).

### 4.2. Diagnostic Skeletal Histology and Imaging 

A variety of imaging modalities can be employed in vivo to image bone metastases. Conventional X-ray radiography provides a rapid and simple means to identify osteolytic/osteoblastic lesions across the animal skeleton. Subsequent 2-dimensional radiographs can be imported to bone and tissue measurement software to quantify lesion area. Using micro-computed tomography to reconstruct bone lesions in rodents or small animals, which is also visually impressive. The benefits of this technique include high-resolution volumetric quantitation of bone microarchitecture. Shortcomings of µCT quantification reside in the current lack of specificity in identifying malignant tissue or quantifying irregular bone surface characteristics derived from metastatic tumor involvement. Bioluminescence provides semi-quantitative visualization regionally indicating sites and the dimensional estimate of metastases following bolus injection or fluorescent agents, such as luciferin or fluorescent antibody, despite exhibiting low spatial resolution. This method helps with preclinical tumor detection and progression in bone and soft tissues.

Ex vivo analysis of bone metastases differs from those performed in vivo. The gold standard for preclinical analysis of tumors and bone is histological analysis. Fluorescent bone labeling using Ca^2+^-binding probes in vivo aids in quantifying dynamic bone parameters such as bone formation rate and mineral apposition rate on plastic-embedded bone sections, however this is not typically employed in the setting of bone metastasis models. These two techniques provide extensive insight into the bone’s metabolic activity when combined with tartrate-resistant acid phosphatase (TRAP) staining to detect osteoclast and resorptive surface. Along with standard hematoxylin and eosin (H&E) staining to quantify various bone and tumor cells, Von Kossa–MacNeil staining permits visualization of osteoblast, lining cell, and osteoid composition on plastic-embedded bone sections. A deeper visualization of osteocyte lacunar-canalicular channels can be performed with SEM on acid-etched bone samples. Together, these techniques provide functional and quantitative insight into critical cellular activity in the marrow and at bone surfaces (Figure 1).

Translating to the clinic, many of the previously discussed techniques remain efficacious, though scaled to meet patient demand. The clinical gold standard for diagnostic imaging to visualize bone metastases undoubtedly requires standard radiology guidance to visualize overtly osteolytic or sclerotic (osteoblastic) lesions. Skeletal scintigraphy involves low-level trace radioactive technetium and X-ray imaging to collocate metabolically active sites where metastases would be evident in bone [66]. X-ray computed tomography advantages reside in the technology’s quantitative capabilities, as changes in bone structure and quality can be tracked longitudinally. Dual-energy X-ray absorptiometry (DXA) detects the changes in bone-mineral density, used to monitor disease progression and treatment effects; however, the resolution utilized in DXA for small animals is far too low to quantify, let alone resolve, metastatic bone lesions. Magnetic resonance imaging is advantageous due to the minimal risk to the patient and pronounced resolution of disease in the marrow being detectable before bone lesions are visible. 

Additional modalities aid in visualizing and quantitating tumor metabolism involving nuclear medicine. Single-photon-emission computed tomography (SPECT) operates similarly to CT (cross-sectional images for 3D rendering), incorporating technetium-99 into the analysis. In contrast, positron emission tomography (PET) improves on the former technique with the injection of the tracer 18F-fluorodeoxyglucose (^18^F-FDG), which has a longer half-life than technetium and performs with much greater spatial resolution. Rapid uptake of glucose by cancer cells enables radiographic imaging of diffuse tumors; a valuable tool in detecting the spread of metastatic bone disease. Combining each of these modalities, such as SPECT/CT or PET/CT [55], has provided invaluable detection capabilities and performance using bone-specific tracers alongside high-resolution 3D imaging. These techniques have vastly improved treatment detection, accuracy, and longitudinal evaluation of lesions in response to treatment (Figure 1). 

## 5. Treatment Approaches for Bone Metastases

### 5.1. Surgical Interventions

The most common sites of bone metastasis in the axial skeleton appear in the vertebrae, ribs and skull, while most appendicular metastases present in the pelvic bone and femora. Surgical interventions (*en bloc* resection, amputation, spondylectomy, etc.) provide an invasive, blunt means of excising cancerous tissue, however, due to the nature of the procedures, they pose a high-risk and can be quite complex in their approach. A coordinated multidisciplinary effort is critical to successful implementation [67]. Deciding when and to what degree to surgically intervene should be considered, depending on patient prognosis. This can be graded using Capanna Class structuring; considerations include solitary and resectable, or if the disease has induced heightened bone pain or a pathological fracture [68]. (Appendix A).

Vertebral metastases can result in extreme pain, mechanical destabilization and compression fracture, which could lead to paralysis; therefore, skeletal metastases must be addressed prudently [69]. Assessing prognosis before treatment in patients with bone metastases, especially in the spine, is critical to achieving the best outcomes. A range of scoring systems have been developed that consider an array of prognostic factors over others, generating scores that reflect patient survival and whether surgery is a worthwhile pursuit. Firstly, the Tokuhashi score [70] addresses the feasibility of surgery using the most recent modification of the score [71] by considering six factors, which include general condition, both the number of foci of extraspinal bone metastases as well as, separately, the number of metastases within the vertebral body, whether or not metastases to other organs exist and, if so, whether they can be removed, the site of the primary cancer, and the degree of spinal palsy, if any exists. This was reported to be the most practical and accurate method with respect to survival in patients with spinal metastases from hepatocellular carcinoma [72], in contrast to other scoring methods. Another approach used alongside the Tokuhashi score is the Tomita method [73], which incorporates a Cox hazard analysis and evaluates the patient based on primary tumor growth as a function of the originating site, the treatability of visceral metastases and whether bone metastases are solitary or contain multiple foci per site. A drawback to this approach, though, is that pain and paralysis are left unaccounted for [74], perhaps inviting more intolerable surgeries than the patient can withstand. In order to simplify this, the one-year survival predictability predicted in the Baur scoring system [75] considers the site of the primary tumor, whether the skeletal metastasis is solitary, and if visceral metastases, lung cancer and pathologic fracture are absent. Despite limitations based on the inclusion of fracture, a score of three to four can indicate a 28.4-month period of overall survival following surgical intervention. Utilizing Karnofsky’s performance measures, the presence of visceral metastases and an individualized point structure based on the primary tumor type, the Linden score [75] suggests an overall survival of 18.3 months following surgical intervention. None of the methods above took into account radiation therapy for spinal cord compression or ambulation, so a series of Rades scoring systems based on an initial system [76] incorporated these factors into scoring systems based on the cancer type: prostate cancer bone metastases [77], breast cancer bone metastases [78] and one for unknown primary lesions [79]. These scores only take radiation therapy into account if it is the singular therapeutic option that has been employed and applying more functional measures of assessment. Lastly, the Katagiri score [80] takes into consideration the bone metastases observed throughout the entire skeleton, as well as the primary tumor based on growth rates, chemotherapy history, performance status and whether there are multiple sites of metastases. This system suffers from objective measuring of individual cancer types. 

Surgical removal of vertebral bone is typically followed by spinal decompression (laminectomy) with plate fixation to relieve pain, yet is not advised beyond a 48 h post-paralysis window [81]. Complete spondylectomy may prove useful pending long-term survival expectations in patients with a solitary lesion [73]. Decompression surgery and radiation are standard treatments of collapsed vertebrae to prevent paralysis, improve pain and restore mobility. Risks associated with spinal decompression surgery include infection, blood clots, adverse reactions to anesthesia, and, in rare instances, death. This procedure is typically followed by radiotherapy post-decompression, as the converse sequence of treatment has been shown to adversely affect wound closure [82]. 

Similarly, pelvic bone incurs high mechanical stresses; therefore, metastases to this site can further perpetuate fracture incidence or severity [83]. Whole or partial arthroplasty is performed, depending on the metastasis location in the four zones across (Enneking Classification) of the iliac crest, to return structure and function to articular joints and surrounding bone [83]. To restore bone tissue lost to resection, needle injections of cement or kyphoplasty followed by cementation are performed. As fractures secondary to bone metastases correlate with reduced patient survival [84], intramedullary nailing has been shown to be an effective technique when coupled with cementation to ensure proper stabilization of the bone [85,86] and for palliation [87]. Radiotherapy, commonly used in palliative care, is commonly used around the periphery of the nail and has been shown to improve overall survival [88]. However, fixtures can fail, and this is speculated to be a consequence of poor implant selection and avoidance of fracture sites associated with radiation [89]. Guidelines based on different criteria (i.e., Mirels and others [90,91,92]) have been issued from numerous radiology societies [93,94,95,96]; for example, the American Society for Radiation Oncology indicates radiation therapy following surgical decompression from spinal cord compression [97]. 

Amputation of the limb distal to the tumor site in a long bone is an invasive strategy that ultimately results in impaired mobility and need for prostheses. The negative impact of these interventions on mobility and quality-of-life is drastically increased; however, the risk of recurring tumor and localized pain is attenuated. Surgical approaches are met with less success the more profuse and invasive the disease. Limb salvage techniques, such as endoprosthetic reconstruction and allografts, have demonstrated improvements comparable to amputation, significantly improving functionality. Endoprosthetic reconstructions, such as the Harrington reconstruction of destroyed acetabulum [98], have become a standard in [99] limb salvage; however, incidence of infections have plagued long-term success rates, at times leading to eventual amputation [100]. Considering these limitations, alternative approaches are sought to spare limb loss and post-surgical complications [101] (Figure 2A), if possible.

One of the alternative strategies to preserve bone while limiting tumor growth is blocking synthesis of tumor-promoting factors. Hormone deprivation is critical in managing the escalation of breast and prostate cancers sensitive to estrogen and testosterone, respectively, to fuel disease progression. ER^+^ breast cancer patients may undergo ovariectomy to remove the dominant source of estrogen. Ovariectomy drastically diminishes circulating estrogen when coupled with aromatase inhibitors. In much the same fashion, bilateral orchiectomy via testicular resection is an effective means of suppressing circulating testosterone in prostate cancer. Accordingly, tumor progression in patients with metastatic bone disease benefits from hormone deprivation therapy by inhibiting tumor progression. Unfortunately, suppressing estrogen synthesis or any of its corresponding substrates (β-estradiol, testosterone), especially for a pre-menopausal patient, critically hinders regulation of the bone remodeling process and its dependency on estrogen to control osteoclast activity. Inadequate management of osteoclast activity can facilitate excessive bone resorption at the expense of bone formation, thus, contributing to aberrant bone remodeling (Figure 2A).

In exceptional cases, after high-dose chemotherapy, breast cancer bone metastases have been treated by autologous bone marrow or hematopoietic stem cell transplantation (rescue). Historical evidence lacked significant increases in overall survival; however, purification of unique cancer biomarkers within the transplantation pool increased efficacy, improving disease-free survival. Surgical resection of bone metastases is performed, although not without consideration to the patient’s overall benefit, and this appears to depend heavily on the site of metastasis. For example, documented cases have shown positive outcomes in patients bearing pelvic metastases [83,102], while patients with breast cancer bone metastases do not necessarily benefit from surgical intervention [103]. Surgery is often indicated when metastases have degraded vertebrae to the extent of causing spinal cord compression (see Section 5.1 on Surgical Intervention) [81] (Figure 2A).

### 5.2. Chemotherapy

Rapid cell division is one of the hallmarks of cancer. By targeting processes that govern cell replication and subsequent separation using chemotherapy drugs in combination, irregularities in division and growth can be introduced that lead to tumor cell apoptosis. Varied cell cycle elements have been used as potential anti-tumor targets. Anti-metabolites (i.e., methotrexate, fluorouracil) are one such class, non-biologic chemotherapy drugs that are frequently used in the cancer setting. Methotrexate has been an important anti-rheumatic drug for rheumatoid arthritis [104], especially when taken in combination with 5-fluorouracil [105]. Repurposed, methotrexate drives apoptosis in bone metastases (and osteosarcoma) through inhibition of dihydrofolate reductase, an enzyme required for nucleotide synthesis [106]. The methotrexate and 5-fluorouracil sequential treatment is also effective in combination with radiation when treating gastric cancer bone metastasis [107,108]. Though commonly used in the setting of cancer, methotrexate alters bone metabolism [109] by decreasing human osteoblast proliferation [110] and contributing to bone defects in cancer patients [111,112]. Associated with profound osteopathy, methotrexate decreases bone formation through defective osteocalcin and subsequent matrix production [113]. Perhaps more alarming are the 25–30-fold increases in IL-6 and IL-11 that, accompanied by increased TNF-α, induce caspase-3 mediated osteocyte apoptosis as well as increase osteoclast activity [114]. Use in childhood cancers renders bone growth arrest by thinning of the growth plate and apoptosis of its chondrocyte constituents [109] (Figure 2B). 

Taxanes (paclitaxel, docetaxel, cabazitaxel) are cytostatic chemotherapy drugs that specifically target and exploit the highly proliferative nature of cancer cells, stalling cytoskeletal microtubule assembly at the G2- or M-phase by binding the spindle fibers together [115]. Mechanistically, aggregate tubulin assemblies required for spindle formation are bound together at the α- and β-subunits to different degrees (depending on the drug or target), rendering them unable to separate following metaphase, leading to mitotic arrest, stalled cell division, and ultimately resulting in cell death. Taxanes are used primarily in addressing breast cancer bone metastases. However, they are approved as only a second-line therapy (docetaxel) for prostate cancer bone metastases [116,117,118] once resistance to androgen deprivation therapy is reached [119] (Figure 2B).

Administration of taxane-class chemotherapeutics is limited by a drug’s absolute toxicity, which is singularly designed to induce cell apoptosis, particularly in cells with heightened replicative machinery. However, a shortcoming of this chemotherapeutic approach against bone metastases is that they target rapidly dividing cells. Therefore, while the primary tumor can most likely be addressed using chemotherapy, circulating tumor cells hidden in the bone marrow niche’s protective environment, rendering a quiescent phenotype, escape the chemotherapy’s mechanism-of-action. Adverse secondary effects of chemotherapy can directly or indirectly impact hematopoiesis and bone metabolism [120]. Specifically, while no changes in osteoblast numbers are observed, as RANKL expression remains unchanged, docetaxel use is associated with decreased bone resorption as a product of reduced osteoclast formation and activity [121]. This results from docetaxel’s direct effects on either early-stage, uncommitted bone marrow-derived macrophages, or inactive multi-nucleated osteoclasts just prior to activation [121]. In contrast, paclitaxel has not been shown to inhibit resorption but instead may induce premature menopause, and this side effect derives from toxicity to the ovaries, thereby interrupting estrogen synthesis. Common side effects shared between chemotherapeutic drugs include dose-limiting toxicity on the bone marrow, which are especially concerning for the hematopoietic system, as anemia is a common side-effect of treatment and initial provocation by the tumor [122], amongst other deficiencies (Figure 2B).

Alkylating chemotherapy agents (doxorubicin, melphalan, cisplatin, cyclophosphamide) are used to target breast cancer bone metastases and castration-resistant metastatic prostate cancer. These agents work on similar principles to taxanes, binding components of rapidly dividing cells. Whereas taxanes bind mitotic spindle fibers; alkylating agents target the dividing nucleic acids. Predominantly sulfur-containing compounds facilitate interstrand crosslinks across guanine and cytosine nucleotides. Cisplatin, a platinum-containing compound, confers similar binding properties to intercalate DNA double strands. Novel approaches have utilized nanoparticles to co-deliver cisplatin with zoledronic acid to treat breast cancer metastases, decrease osteoclast activity and reduce osteolysis [123]. These strong bonds prevent topoisomerases from separating the dividing DNA complex. Another unique example is cyclophosphamide’s cyclization upon binding to the opposing guanine nucleic acids, which aids in prostate cancer treatment. Evidence has pointed to earlier time points in cancer’s progression being more advantageous to successful treatment outcomes (Figure 2B).

Unfortunately, the drawbacks to high-dose chemotherapy with alkylating agents also lead to dysregulation in bone remodeling, decreasing bone formation by preventing PTH from binding to its osteoblast receptor and upregulating bone resorption by damaging the ovaries. Cisplatin upregulates bone resorption while downregulating bone formation, thereby saturating the body with calcium to produce renal toxicity. Cyclophosphamide has profound adverse effects on bone, arresting both pre-osteoblast and pre-osteoclasts, to the degree at which no remodeling cells remain. Ultimately, defective bone mineralization is compounded by toxic effects on the ovaries, which contributes to hypogonadism leading to premature menopause, and kidney failure, as observed with ifosfamide (Figure 2B). 

Proteasome inhibitors (PI) are a unique class of chemotherapeutic agents that target rapid metabolism and excess accumulation of extra-lysosomal protein and/or waste material by blocking the ubiquitin-proteasome pathway [124]. Due to this highly-conserved pathway, bortezomib has exhibited significant clinical efficacy and has become an invaluable FDA-approved agent in treating relapsed multiple myeloma [125,126]. However, as bortezomib has shown effects against anti-endocrine therapy-resistant ER^+^ breast cancer and promotes bone formation, mediated by increasing BMP-2 [127], it has proven helpful in metastatic breast cancer bone disease [128,129,130]. Promising preclinical evidence for treating prostate cancer bone metastases indicates that the utility of bortezomib may extend beyond hematological malignancies in bone and breast cancer bone metastases [131,132]. Additionally, bortezomib decreases osteoclast activity through a reduction in the nuclear factor-kappa-beta (NF-κβ) and osteoclast differentiation [133]. While the positive effects outweigh the negative, toxicity over prolonged treatment has surfaced, necessitating a next-generation PI, carfilzomib, which has a lower neurotoxicity profile. Nevertheless, even carfilzomib exhibits adverse effects associated with renal toxicity, and perhaps more alarming, cardiovascular toxicity [134]. 

### 5.3. Anti-Resorptive Agents for Bone

Bisphosphonates are a family of anti-resorptive drugs that chemically resemble inorganic pyrophosphate, thus, exhibiting a strong binding capacity to hydroxyapatite crystals in bone [135]. The ability of these drugs to suppress bone resorption through a potent negatively regulating osteoclast activity have reinforced their prescription as a mainstay of clinical use in addressing imbalances in bone metabolism for nearly 50 years. Iterations of bisphosphonates have aided in inhibiting osteoclast activity through uptake mechanisms that block the activity of pathways critical to osteoclast structure assembly and survival, subsequently resulting in osteoclast apoptosis and inhibition of osteocyte apoptosis. Bisphosphonates harbor the paradox of a treatment working too well; nearly all bone resorption is inhibited, so bone turnover is not normalized despite bone retention, potentially increasing fracture risk. However, inhibition of bone metastasis progression has been shown following bisphosphonate administration. Oral bisphosphonate also effectively reduces breast cancer tumor burden [136]. This suppressive effect on proliferation is partially derived from its interaction with the bone marrow microenvironment; inhibited osteoclast activity prevents the release of bone-derived inflammatory factors that perpetuate tumor growth [137]. Unfortunately, prolonged treatment with bisphosphonates can produce undesired toxicities. Adverse reactions have been cited following long-term use, leading to atypical fractures or osteonecrosis of the jaw (ONJ). 

Drugs inhibiting the osteoclasts activation are also in use. The anti-resorptive drug denosumab is the fully human monoclonal antibody to RANKL [138]. Denosumab’s initial use was directed towards the prevention of SREs related to primary breast and prostate cancers and in multiple myeloma progression [139], a bone cancer highly dependent on the influence of RANKL in upregulating osteoclast-mediated resorption and myeloma cell growth. Indeed, inhibition of RANKL blocks osteoclast maturation and prevents subsequent bone resorption resulting in slowed bone metastases progression while reducing tumor-associated osteolysis, as demonstrated in breast, lung and, prostate cancer metastases [140,141]. Denosumab has been shown to be superior to bisphosphonates (zoledronic acid) with regards to offsetting SREs [142], exhibiting a longer half-life, specifically in the prevention of hypercalcemia of malignancy in bone metastasis patients [143]. However, in early-stage breast cancer metastasis patients, denosumab conferred no specific advantage in reducing breast cancer recurrence or minimizing death [144]. Additionally, immediate cessation of denosumab in patients has led to cases of rebounding hypercalcemia and heightened bone remodeling following drug holidays or discontinuation [145,146]. This area requires attention and vigilance of fluctuating calcium levels, which can lead to skeletal fractures [147]. Further, denosumab use in children is sensitive and can cause ONJ in rare cases [148,149] (Figure 2C). 

### 5.4. Anabolic Agents for Bone

Bone metastases are characterized by a high degree of abnormal bone remodeling, most often consisting of osteolytic lesions. Restoration of the bone lost due to heightened resorption is rarely achieved. Osteoanabolic (i.e., teriparatide, abaloparatide) drugs were developed to induce PTH-driven osteoblast-derived bone formation in osteoporotic women [150,151]; however, black-box warnings have prevented the use of this in the cancer setting, citing concerns over the induction of sarcomas [152]. The FDA lifted this warning in 2020, once again opening the potential for such approaches to increase bone formation in metastatic bone disease. 

Sclerostin, a potent inhibitor of the Wnt signaling pathway produced by mature osteocytes, is a negative regulator of bone formation in healthy bone remodeling [153], presenting as a potential and widely expressed target for bone formation. The antibody to sclerostin (romosozumab) was recently approved by the FDA to treat bone healing in osteoporotic women at high risk of fracture [154]. Studies are now exploring romosozumab as a pharmacological candidate to alleviate bone breast cancer bone metastases [155]. Nevertheless, adverse effects have been cited in other studies using romosozumab, prompting careful consideration of its implementation in the clinical setting [156,157] (Figure 2C).

### 5.5. Small Molecular Bone Targets

The significant role of the growth factor TGF-β in the initiation, perpetuation, and treatment of cancers cannot be understated, as alterations in its pathways play a tumor-suppressive role [158] at early stages and a promotor during more progressed states [159]. This emphasis is founded on the many functions TGF-β regulates in normal tissue [160]. The enrichment of TGF-β embedded within the bone matrix makes it a critical regulator of osteogenesis and a unique driver of the metastatic process [161,162], as are BMPs and Activin. Outside the loss- or gain-of-function effects, TGF-β also suppresses the cytotoxicity of CD8 effector cells [163], in turn favoring osteoclast activity. However, due to its extensive and complex interactions with other molecules, cells, substrates, and its superclass members, TGF-β is a distinct target with therapeutic potential [164]. TGF-β binds to surface receptor kinases that initiate phosphorylation of downstream SMADs that translocate to the nucleus [165]. As such, receptor-tyrosine kinases have been the target of many therapeutic efforts to minimize tumor burden in different cancer types [166]. Here, inhibiting the TGF-β receptor kinase-1 type-I receptor kinase and SMAD proteins has led to varied outcomes in preclinical and clinical studies (Figure 2D).

### 5.6. Radioisotope Therapy 

Treating osteolytic lesions necessitates the use anti-resorptive agents as gold-standards to stem bone resorption; however, the osteoblastic and mixed lesion phenotypes respond to different approaches. Radionuclides consisting of either α-, β-, or γ-emitting isotopes have been demonstrated over the past decade as effective in suppressing osteoblastic bone metastatic cancer, such as advanced prostate cancer bone metastases [167,168]. Strontium (Sr^89^) isotopes are analogs to Ca^2+^ [169]; thus, they are readily taken up by active osteoblasts and embedded in the bone matrix. β-emission from Sr^89^ targets the nearby malignant osteoblasts driving osteosclerosis. The affinity of Samarium (Sm^153^) to bone is derived from co-delivery with bisphosphonate molecules, which bind to hydroxyapatite surfaces [170]. Once embedded in the bone matrix, β-emission induces tumor cell apoptosis while γ-emission aids in diagnostic imaging. Both techniques equate to ~75% response rates. Radium-223 (Ra^223^) is a targeted α-therapy for castrate-resistant prostate cancer bone metastases due to its survival benefits [171]. Ra^223^ preferentially incorporates into the newly formed bone in osteoblastic bone metastases. It emits α-particles that break double-stranded DNA in tumor and bone cells to promote tumor cell death; thus, Ra^223^ interrupts the dynamic interaction between the tumor and the bone microenvironment [172]. Ra^223^ has also shown promise in advanced breast cancer bone metastases patients, improving both bone and tumor endpoints; however, further analysis will determine the long-term benefits to patients [173] (Figure 2D). 

### 5.7. Radiation Therapy

Radiation is a powerful therapeutic tool commonly used to manage advanced and metastatic diseases [174]. External X-ray beam irradiation on bone lesions allows for precise ablation of malignant tissue that is difficult to reach using conventional surgery [175]. Intensity-modulated radiotherapy can improve this level of accuracy by using computed guidance to alter beam trajectory, especially for controversial sites such as vertebral metastases [176]. Brachytherapy adopts the approach of implanting a sealed vehicle containing radioactive material that can be placed near tumor cells without the invasiveness of surgery or damage derived from high-energy external beam irradiation. Proton therapy is used in combination with chemotherapy drugs to treat bone metastases bone [177] without the drawbacks of high-energy radiation. The adverse effect of radiation on bone and marrow health is well-studied [178,179,180] compromising bone mineralization and the stem cell pools critical to maintaining bone health. Indirect effects on bone have been attributed to proton therapy targeting prostate cancer, increasing the incidence of hip fractures [181]. Radiotherapy can also be used as palliative care for pain management in advanced cancers when no alternative treatment option is available. Fractionated doses (i.e., 6 × 4 Gy/ea) have been considered more tolerable for the patient and exhibit greater efficacy than single-fraction doses at 8Gy, for example, recent studies highlight equal palliation and survival under the latter strategy, providing a more economical and tolerable approach for the patient [182,183,184].

### 5.8. Pharmacological Developments and Novel Immunotherapeutic Targets 

Cytotoxic lymphocytes secrete granzymes to target malignant cells [185]; therefore, blockade of immune checkpoints has risen to prominence as a major therapeutic strategy [186]. Researchers have understood the mechanisms and circumstances surrounding immune checkpoints; however, it was not until the late 1990s that James Allison led the discovery of the CD28 homolog cytotoxic-T lymphocyte associated protein-4 (CLTA-4) known as CD152, an immunosuppressive checkpoint [187] that researchers have since harnessed as a tool to combat disease. By inhibiting CTLA-4 (drug name ipilimumab), the costimulatory complex CD28:B7-1/2 remains unbound, allowing T cells to bypass the inactivation of naïve T cells to differentiate into lineage-specific CD4, helper- (Th) and regulatory-T (Treg) cells. In turn, highly-activated CD4 cells are primed to seek out malignant cells. The novelty of this approach might lie with the emergence of a new CD4 phenotype, one that persists post-treatment. Additionally, CTLA-4 has been demonstrated to increase bone remodeling [188]. Since then, other antibodies to T cells have emerged, notably to programmed death-1 (i.e., PD-1). PD-1 inhibition via pembrolizumab (also nivolumab) is now a first-line treatment, particularly in cancers unreachable by conventional surgery. As with nearly any therapy, toxicities surface over time, and some of these are quite severe. Inflammatory bowel syndrome, lung inflammation, and other more common, although less alarming, side effects occur (Figure 2F). 

More recent advents in the immunotherapy space include chimeric antigen receptor-modified (CAR-T) T cell technology. In short, autologous T cells are engineered to express CD19 antigen to target tumor cells. Initially targeting leukemias without bone marrow transplantation has led to studies targeting solid tumors; however, infiltrating bone metastases with CAR-T therapy has been met with challenges. Known ligands C-X-C chemokine receptor type-4 (CXCR4) [189] and C-C motif chemokine ligand 2 (CCL2) [190] expressed in prostate cancers are being introduced to CAR-T cells to ramp up T cell trafficking to the tumor. In similar efforts, IL-12 has been engineered into CAR-T cells targeting tumor stroma [191] and to mount macrophage responses [192] on tumor tissue. Transmembrane AMPA receptor regulatory protein (TARP) containing CAR-T cells have been utilized in clinical studies as a possible tool in targeting both breast and prostate cancer metastases [193]. Very recent work on mesothelin, an overexpressed marker in breast cancer [194,195], targeted by CAR-T cells, was shown to inhibit breast cancer growth and be enhanced with the administration of an oncolytic adenovirus that targets TGF-β [196,197]. As additional candidate markers become unveiled, other metastatic and non-metastatic cancers will be targeted using engineered T cell technology (Figure 2F). 

## 6. Alternative Approaches

### 6.1. Combinatorial Therapies

As therapies and clinical data on the use of complex therapies become increasingly available for varied populations, new and intricate approaches are being considered. Tumor heterogeneity presents a problematic hurdle to address, as the target may not an exclusive phenotype, rather many, and could respond differently depending on the chosen therapy or evade drug therapy altogether. For instance, in prostate cancer bone metastases CTLA-4 is highly evident but infiltrating T cells are scarce; therefore, anti-CTLA-4 (ipilimumab) is one such drug strategy that is being used to revitalize immune cell attraction to the tumor. However, when treated with ipilimumab and PD-1, prostate cancer cells in the bone bypassed drug-induced immune attacks, presenting with upregulated bone destruction driven by TGF-β-induced tumor cell growth while also blocking activated T cells’ effects [198]. Conversely, ongoing efforts are evaluating the efficacy of combining CTLA-4 and TGF-β inhibitors against castrate-resistant prostate cancer in mice, citing decreased bone metastasis growth and T regs, while increasing Th1; findings the researchers expect to advance into clinical application. Many regulatory processes exploited by cancer cells overlap with those of immune cells [199]. Increased understanding of compensatory mechanisms associated with tumor and bone are generating more complex strategies to approach each disease. 

### 6.2. Senescence and Senolytics

The concept of senescence, natural aging, and cell-cycle arrest has been the focus of increased research in recent years, especially considering senescent cells’ effects on aging have been identified as responsible for disease onset. Due to their potent effects on cell cycle arrest, senolytic compounds have been explored as potential anti-cancer agents [200]. Heat-shock protein-9 (HSP-9)-inhibitor, AKT-inhibitor, or atraric acid have been used to induce tumor cell senescence. Galangial leaf extracts have been cited as possible senescence-inducing compounds that could exhibit additive effects in combination with doxorubicin by inhibiting metastatic breast cancer cell migration and growth [201]. Unfortunately, the senescence-associated secretory phenotype (SASP) has emerged as a troubling obstacle to senolytic implementation [202,203] since an alteration to the bone microenvironment can confer oncogenic activation and loss of tumor suppression [204]. Even senescent non-tumor cells can contribute to aggressive relapse or fall victim to neighboring cells, as paracrine mechanisms [205] have been identified that confer SASP-mediated alterations to the tumor microenvironment [206]. Positive and negative outcomes have been associated with the compounds against singular disease. Even in the fight against prostate cancer, different senolytics exhibit both senescent and some pro-survival signals [207] (Figure 2G). 

### 6.3. Hyperthermia

The thermal and electrical permittivity of bone and the internal marrow is relatively low and inefficient compared to other tissues, making the use of hyperthermic conditions an attractive route to reach an inaccessible tumor [208]. Hyperthermia has been shown to increase bone deposition in vivo [209]. Adequate temperatures are ramped up over a 15 min timespan, lasting approximately 2 h until treatment is complete. Alone, its use has shown selective improvements in bone metastases [210] that have shown resistance to radiotherapies. Application of hyperthermia as a sensitizer of cells in vitro [211,212] and tumors to specific treatments renders the cells more susceptible to radiation [213] and chemotherapies [214,215]. Novel treatment with hyperthermia has been performed using electromagnetic fields to target the bone metastases engendered with magnetic material [216]. Side effects, including pain, thermal burning, and the cutaneous blisters, are the extent of immediate clinical complications, though even these are decreasing with improved technology. General nausea, vomiting, and diarrhea accompany some treatments, though heightened liver and kidney use (following increased treatment uptake) can temporarily produce jaundice. Nevertheless, the use of hyperthermia as an adjunct to established therapies is advantageous in solid, unreachable tumors, highlighting its potential to improve drug-delivery to the site of concern while limiting dosage and post-treatment complications. 

## 7. Palliative Care

A substantial number of patients with metastatic bone disease experience heightened bone pain, amongst other common symptoms, which becomes exceedingly challenging to address and both debilitating and costly for the patient. Various palliative care routes provide relief using anti-cancer agents to suppress tumor burden and analgesics to relieve symptomatic neuropathy. However, data suggest pain management is being inadequately addressed [217]. Dull aches characteristically initiate awareness of the metastatic disease; however, these aches often progress into episodic pain and can accompany bone fractures. Treatment strategies to address tumor burden in metastatic bone can then accelerate pain even further. 

Safe and minimally invasive methods have been used to address local skeletal metastases and/or alleviate bone pain [218], such as radionuclide therapy, radiofrequency ablation [219], focused ultrasound [220], and cryoablation [221], and others that have been previously mentioned. Radiofrequency ablation (RFA) is used to treat bone metastases by directly targeting the tumor and is also effective in relieving pain symptoms [175], yet probes needed to reach the target tissue are limiting factors. Additionally, RFA is contraindicated in spinal metastases that fall in short range (≤1 cm) of the spinal cord [222]. Magnetic resonance-guided focused ultrasound [223] and CT-guided cryoablation [224] to accessible tumor have shown moderate success in reducing pain associated with metastatic bone disease. Primary osteoid osteomas (non-metastatic, benign tumors) are treated in this manner as there is minimal involvement with the underlying marrow in contrast to systemic approaches, lessening the morbidity in healthy tissue. The MOTION clinical trial highlighted the robust palliation of pain in metastatic bone tumors using cryoablation [225]. The completion of the procedure can involve cementation and plate fixture of the bone to provide mechanical resilience and prevent fracture [226]. Complications have been noted, with secondary fractures, infection, and bleeding constituting most reported cases [227,228]. For these reasons, oncologists look to pharmacologic agents to provide additional relief (Figure 2H). 

Alternatively, multi-generational bisphosphonates have shown varied yet significant clinical benefits to relieving bone pain in addition to their known anti-resorptive activity [229]. This prolonged benefit has been observed in many cancers, specifically those necessitating relief from bone pain, such as breast, prostate, and lung metastases [230,231,232,233]. Despite the overwhelming benefit, prolonged use with bisphosphonates generates adverse toxicity to the kidneys and ONJ; the latter, however, constituting a relatively rare clinical event. Analgesics, such as non-steroidal anti-inflammatory drugs (ibuprofen), acetaminophen, and opioids (e.g., fentanyl, morphine, and oxycodone) [234], have been used to address mild pain from bone metastases and in conjunction with narcotic analgesics for more severe pain [235,236]. Corticosteroids, tricyclic antidepressants, anticonvulsants, and neuroleptics are used in combination with opioids for improved pain control [237]. Prescription opioids and narcotics have experienced increasing pushback, as addiction and overuse have surfaced, leading to adverse, long-term complications marked by dependency and, in certain instances, increased patient mortality. Alternatively, medicinal cannabis has been prescribed for patients suffering from physical and mental anguish derived from cancer [238]. Physician concerns over improper training and misuse of cannabis have limited its extensive usage [239]. However, attitudes towards its use and research on its target profile have garnered more favorable attention in recent years. Cannabinoids and terpene compounds extracted from cannabis plants have been proposed as novel analgesics in metastatic bone disease patients. Type-1 and -2 cannabinoid receptors are multifaceted, exhibiting properties that regulate bone [240], cancer, and pain perception [241]. Endocannabinoids applications are shared across these areas; cannabinoid-2 selective ligands and their associated receptors interact, providing pain relief and reducing complications with bone metastases. However, the legality of their use internationally has limited their widespread application in this setting (Figure 2H).

## 8. Gaps in Knowledge 

Cancers derived from breast, prostate, thyroid, renal, and lung origin exhibit large heterogeneity, which complicates the understanding and management of the primary disease and its metastatic progeny. Cells that escape the detection and elimination through any modalities described above may comprise a phenotypic subpopulation that is not distinctly targeted by the treatment modality. Again, treating metastatic tumors of heterogeneous composition constitutes a significant complication facing nearly every patient in the advanced stages of the disease. Single-cell genomic analysis of transcriptomes [242] to elucidate key biomarkers across primary tumors and bone metastases has led to the identification of genes linked to survival, common to primary breast and prostate cancers and metastases to bone [243,244]. Further, the frequency and temporal sequencing of combinatorial treatments is still an ongoing area of research, specifically the time between bone-targeted agents and chemotherapeutics. For instance, retrospective analyses have pointed to improved completion of Ra^223^ if pre-chemotherapy in metastatic castrate-resistant prostate cancer was initiated, although this has not translated into changes in survival outcomes [245]. Ra^223^ treatment in combination with paclitaxel exhibited no toxicities in various cancer metastases to bone, ushering further investigation into its targeted efficacy of bone metastases [246]. Similarly, treatment of solid tumor bone metastases with immunotherapy needs further exploration, as well as the long-term effects on musculoskeletal tissue. The utility of senolytic compounds is far from conclusive, with additional studies needed to circumvent SASP-mediated tumor suppression whilst maintaining durable anti-tumor effects. Overall, a comprehensive understanding of the mechanisms governing the progression of each metastatic bone disease and the individualized systemic and tissue-specific responses to treatments will provide the foresight necessary to calculate the most efficient route to achieving disease remission. 

## 9. Conclusions

Bone metastases remain one of the leading causes of cancer-associated morbidity and mortality. Efforts to implement translational findings to the clinic have been varied. Recent advances have focused on targeting the cellular and molecular elements of bone resorption to stem tumor progression. Challenges in maintaining disease-free survival and eliminating pain persist, despite significant advances and new avenues in the field (Appendix A). The initiation of secondary complications in the wake of available treatments remains a cause for concern. Attention to bone preservation amidst metastatic disease should be a crucial consideration to patient treatment and recovery as well as for novel therapeutic strategies targeting the disease; spare healthy musculoskeletal tissue, if possible, lower the incidence of SREs, and improve quality-of-life.

## Figures and Tables

**Figure 1 cells-11-01309-f001:**
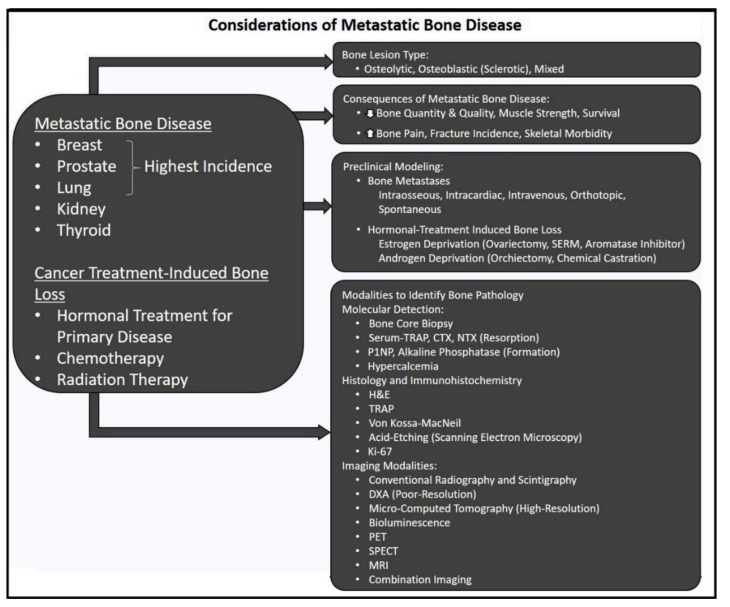
Metastatic bone disease and the various treatment strategies employed to combat tumor progression corrupt the bone remodeling pathway. Incidence and lesion type vary depending on the phenotype of the primary malignancy and can even vary amongst subtypes within bone metastases. Treatments can directly target bone matrix (irradiation) or indirectly (hormone deprivation, chemotherapy) by interrupting pathways critical to maintaining normal bone remodeling and healthy musculoskeletal tissue. Systems modeling metastatic bone disease can be achieved by introducing murine and human cell lines intravenously, left ventricular intracardiac inoculation, intraosseous injection into tibiae or femora, orthotopically or using spontaneous models (predominantly canine). Cancer treatment-derived bone loss can be simulated by modeling hormone deprivation protocols, introduction of pharmacologic agents and under skeletal irradiation. Detection of bone pathology in the laboratory is performed using a variety of histological techniques to identify and quantify tumor burden, the activity and number of bone remodeling cells and the amount of substrate they are acting on and the areal and spatial quantification of bone mineral content. Longitudinal in vivo and typically higher-resolution ex vivo imaging techniques range in resolution, speed and the detection of metastatic tumor progression and treatment response, modalities which are also limited by the animal’s tolerance of a multiple round of anesthesia and the severity of the disease progression.

**Figure 2 cells-11-01309-f002:**
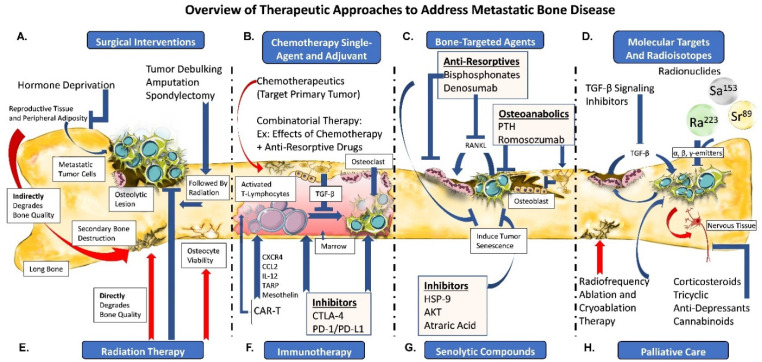
Therapeutic approaches to treat metastatic disease are varied, eliciting anti-tumor properties although not without secondary effects to the surrounding musculoskeletal tissues. (**A**) Surgical removal of diseased tissue typically includes amputation, heightening skeletal morbidity. Pharmacological blockade or surgical removal of tissues that produce sex steroids to limit tumor progression can act negatively on musculoskeletal tissue, as estrogens and testosterones are major regulators of physiological bone remodeling. (**B**) Chemotherapeutic drugs, particularly in combination with bone-targeted agents (**C**) are employed in combination and in varying time schemas to attack tumor in bone while suppressing heightened resorption with anti-resorptive drugs. (**D**) TGF-β pathway inhibitors have been tested for their efficacy in reducing tumor burden in preclinical animal models and clinical trials. Small molecules with radioactive properties, such as radium-223, exhibit dual targeting properties to reduce tumor burden and improve bone health in metastatic bone diseases. (**E**) Many radiation therapies are a means of targeting difficult to reach tumors and as a palliative care approach; however, damage to nearby bone tissue and the underlying stem cell pool can increase fracture risk for patients treated following high-dose radiation. (**F**) The burgeoning field of immunotherapies utilize CAR-T and other antigen-presenting schemas to activate lymphocytes to target tumor cells, while (**G**) senolytic compounds are being explored as a means to induce tumor cell senescence. Patients with metastatic bone disease exhibit a high degree of bone pain; therefore, palliation casts a wide net, with (**H**) interventions ranging from tumor ablation to pharmacological suppression of pain receptors. (Note: Red arrows indicate adverse effect on bone).

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
