# Peer review of "Translational Strategies to Target Metastatic Bone Disease"

_cells, 2022, doi:10.3390/cells11081309_

Round 1

Reviewer 1 Report

Thank you for the opportunity to review the manuscript. This review attempted to take a translational approach to the therapy of bone metastases. The authors have succeeded in providing a good overview of the complex topic and numerous current citations are given. In addition to the theoretical basis for the development of bone metastases, various diagnostic and therapeutic modalities and alternative or experimental therapeutic approaches are described. 

A limitation of the work is that the authors have tried to cover this very complex topic completely and thus could not go into the different tumor entities in depth. Here it would be appreciated if a few references for the reader could be added, so that specific therapies or diagnostic modalities can be read upon. 

Author Response

Reviewer 1:

Thank you for the opportunity to review the manuscript. This review attempted to take a translational approach to the therapy of bone metastases. The authors have succeeded in providing a good overview of the complex topic and numerous current citations are given. In addition to the theoretical basis for the development of bone metastases, various diagnostic and therapeutic modalities and alternative or experimental therapeutic approaches are described.

A limitation of the work is that the authors have tried to cover this very complex topic completely and thus could not go into the different tumor entities in depth. Here it would be appreciated if a few references for the reader could be added, so that specific therapies or diagnostic modalities can be read upon.

Response to Reviewer 1 Question 1: We thank the reviewer for the positive response to the manuscript and appreciate the limitations addressing the scope of this subject in a singular manuscript. For example, we have added additional information and references relating to therapeutic strategies and diagnostic modalities of consideration for thyroid, lung and renal bone metastases, though there is large overlap with breast and prostate cancer bone metastases. Despite this overlap, specific tumor cell sensitivity to varied treatment modalities can hinder rates of success. We hope that these additional references will suffice, considering the scope of the topic.

Reviewer 2 Report

This review deals with a very important subject in the field of oncology, that of bone metastases. In the first chapters, this review presents in a very clear way the clinical characteristics of metastatic bone disease, the main cellular mechanisms that cause bone metastases and the classical treatments of these pathologies (androgen deprivation therapy, estrogen deprivation therapy....).

The review continues with the modelling of metastases by imaging and the description of preclinical models allowing the study of these bone metastases. The review concludes with a presentation of classical therapeutic approaches to bone metastases (surgical interventions, chemotherapy, antiresorptive agents, anabolic agents...) and by the presentation of innovative or experimental therapies (targeting of the TGF-β axis, pharmacological developments and new immunotherapeutic targets, combinatorial therapies, senescence, hyperthermia).

This review is very well written with quality figures and a well-cited bibliography. For ease of reading, a table or tables summarizing conventional and innovative therapeutic strategies would be welcome.

Author Response

Reviewer 2:

This review is very well written with quality figures and a well-cited bibliography. For ease of reading, a table or tables summarizing conventional and innovative therapeutic strategies would be welcome.

Response to Reviewer 2 Question 1: The reviewer’s kind reception of the manuscript is very much appreciated. In addition, we have added a table to summarize both conventional and innovative therapeutic strategies for ease of reading and coupled with their known impact on bone. This very concept was meant to be conveyed illustratively in Figure 2, but we agree it is highly-beneficial to provide a succinct tabular representation of the different approaches mentioned in this paper. We hope the inclusion of this table with address the reviewer’s request.

Reviewer 3 Report

The review described the treatment of metastatic bone tumors.

1) The title is "to target metastatic bone disease". But a large part of this review is limited to prostate and breast cancer. Recently, we have often seen patients with bone metastasis from lung cancer and renal cancer.

2) The authors picked up the hormone deprivation therapy for cancer-treatment induced bone loss. However, some cytotoxic agents (e.g. MTX) can induce it.

3) I strongly recommend the modification of the surgical intervention paragraph. The authors should review the indication of surgical intervention. I think surgical interventions should be considered if the bone metastasis is solitary and resectable, pathological fractures, and painful at a long bone. 

Also, the authors should discuss spinal metastasis.

4) Line 515-517 Please add the references. I don't know the superiority of single-dose fraction for bone metastasis, compare to several fractions.

5) Ref 127 is a clinical trial announcement. It should be appropriate as a reference of Line 520-521.

Author Response

Reviewer 3:

  1. The Title is “to target metastatic bone disease”. But a large part of this review is limited to prostate and breast cancer. Recently, we have often seen patients with bone metastasis from lung cancer and renal cancer.

Response to Reviewer 3 Question 1:  Thanks to the reviewers for presenting this point. We agree the title is indicative of overall metastatic bone disease and that the body of the text largely addresses metastatic bone disease in the context of breast and prostate cancer bone metastases, as they are the largest contributing cohorts. Likewise, we agree that other cohorts derived initially from thyroid, renal and lung cancers, although to a lesser extent, can progress into bone metastases. As such, we have added more information regarding thyroid, renal and lung cancers as it pertains to each of the sections covered in the manuscript. It should be noted that many of the references already included pertained to studies that addressed bone metastases in general, and not just an exclusive study of breast and/or prostate cancer bone metastases. Nevertheless, we have made specific reference to bone metastases of other origin (lung, kidney, gastric carcinomas).

  1. The Authors picked up the hormone deprivation therapy for cancer-treatment induced bone loss. However, some cytotoxic agents (e.g. MTX) can induce it.

Response to Reviewer 3 Question 2: We thank the author for their suggestion. Methotrexate is a commonly used chemotherapy drug and its use, with or without 5FU is an effective approach to treating bone metastases. Methotrexate was originally mentioned in the manuscript alongside taxane class chemotherapeutics; however, we have now moved the entirety of this description (including mechanism of action and the side effects on bone in the setting of cancer) to the first paragraph in the respective section “chemotherapy”. Additional references to methotrexate use in metastatic cancer has also been incorporated into the discussion.

  1. I strongly recommend the modification of the surgical intervention paragraph. The authors should review the indication of surgical intervention. I think surgical interventions should be considered if the bone metastasis is solitary and resectable, pathological fracture, and painful at a long bone. Also, the authors should discuss spinal metastasis.

Response to Reviewer 3 Question 3: We truly appreciate this guidance, as it is critical to be precise when understanding the qualifications for surgical intervention. Considering the reviewer’s suggestions, we have adjusted our responses to include surgical intervention if the bone metastasis is solitary and resectable, consists of pathological fracture, and painful at a long bone. Additionally, we have included a discussion on spinal metastases and the surgical intervention process for this clinical challenge along with a discussion on the surgical treatment of pelvic metastases. Capanni and Enneking classifications are discussed as well, to highlight to the reader that surgical interventions are guided by location and severity of the metastases in the individual bone compartments.

  1. Line 515-517 Please add the references. I don’t know the superiority of single-dose fraction for bone metastasis, compare to several fractions.

Response to Reviewer 3 Question 4: Thank you, for pointing this out, as this is an important consideration in the compromise between treatment efficacy and practicality. Single-dose radiation is the suggested standard in palliative care of bone metastases, as there is no evidence that the multi-fractionated doses are advantageous. As such, we have reworded our statement to reflect this point and have added multiple references, as requested.

  1. Ref 127 is a clinical trial announcement. It should be appropriate as a reference of Line 520-521.

Response to Reviewer 3 Question 5: We are sorry for the confusion. Ref 127 was submitted in the first draft as a reference in Line 520-521. If the comment mentioned by the reviewer is a typographical error and meant to convey that REF 127, a clinical trial announcement (we don’t believe it to be a clinical trial announcement), is inappropriate for Lines 520-521, then we will remove it from the manuscript.  

Round 2

Reviewer 3 Report

Thank you for submitting the revised manuscript. I think surgical intervention should be considered depending on the prognosis of the patients. Tokuhashi score, Tomita score, Baur score, Linden score, Rades score, and Katagiri score were used for predicting survival and finding the merit of surgery. The authors should describe those. The authors don't discuss the relationship between the prognosis and the indication of the surgery. Also, the authors should discuss the combination therapy of surgery and radiotherapy. Usually, spinal decompression is performed and after surgery, radiotherapy should be considered. Also, radiotherapy should be considered around the intramedullary nail for a pathological fracture.
